# Identification of Gait Motion Patterns Using Wearable Inertial Sensor Network

**DOI:** 10.3390/s19225024

**Published:** 2019-11-18

**Authors:** Kee S. Moon, Sung Q Lee, Yusuf Ozturk, Apoorva Gaidhani, Jeremiah A. Cox

**Affiliations:** 1Department of Mechanical Engineering, San Diego State University, 5500 Campanile Drive, San Diego, CA 92182, USA; apoorva2501@gmail.com (A.G.); cox_j_a@hotmail.com (J.A.C.); 2Electronics and Telecommunications Research Institute, ICT, 218 Gajeong-ro, Yuseong-gu, Daejeon 34129, Korea; 3Department of Electrical and Computer Engineering, San Diego State University, 5500 Campanile Drive, San Diego, CA 92182, USA; yozturk@sdsu.edu

**Keywords:** gait analysis, wearable sensors, inertial measurement unit, human kinematics, phase difference angle

## Abstract

Gait signifies the walking pattern of an individual. It may be normal or abnormal, depending on the health condition of the individual. This paper considers the development of a gait sensor network system that uses a pair of wireless inertial measurement unit (IMU) sensors to monitor the gait cycle of a user. The sensor information is used for determining the normality of movement of the leg. The sensor system places the IMU sensors on one of the legs to extract the three-dimensional angular motions of the hip and knee joints while walking. The wearable sensor is custom-made at San Diego State University with wireless data transmission capability. The system enables the user to collect gait data at any site, including in a non-laboratory environment. The paper also presents the mathematical calculations to decompose movements experienced by a pair of IMUs into individual and relative three directional hip and knee joint motions. Further, a new approach of gait pattern classification based on the phase difference angles between hip and knee joints is presented. The experimental results show a potential application of the classification method in the areas of smart detection of abnormal gait patterns.

## 1. Introduction

Early detection of symptoms enables better chances of treatment and sometimes complete cure of many diseases. Many health-related symptoms can be diagnosed in their primitive stages by monitoring body functions continuously. With the advancement in wearable sensor technology, many new pervasive monitoring systems are available in the market that can track physiological signals and wirelessly transmit them [1]. Many issues are related to gait disorders such as asynchronous body movement, asymmetry, loss of smoothness, and slowing of gait speed, etc. The understanding and identification of the human gait have generated much interest in many research fields beyond medical sciences such as biomechanics, etc. For the study of human gait, researchers have used various techniques in lab environments [2].

A gait cycle is described as the sequence and rhythm of foot/limb movements in the walking pattern of a person to achieve the motive of locomotion by propelling the body’s center of gravity forward. A gait cycle comprises of a stance phase and a swing phase. The stance phase corresponds to the instance while walking when the body balances on both the feet, whereas the swing phase is the one in which the body balances on one foot while the other foot is off the ground to take the next step. Normal or healthy individuals have a normal gait, but certain neurological and musculoskeletal disorders cause abnormality in gait pattern [3,4]. Notably, mobility and disability problems, including the gait disorder, tend to increase within the elderly population. Some gait cycle patterns generally change with age, such as gait velocity, double stance time, walking posture, and joint motion [5].

Gait assessment is one of the essential steps when diagnosing neurological conditions, orthopedic problems and medical conditions. [6]. Manifestations of an abnormal gait are the loss of symmetry of motion and timing between the left and right sides, which can cause difficulty in initiating or maintaining pace. The three-dimensional gait analysis enables the detection of angular deflections during gait; this is important for clinical purposes to be able to identify different disorders accurately [7]. A standard method for measuring gait cycle is to use optical tracking systems such as the Vicon motion analysis system. A set of cameras track the markers attached to the human body to measure and calculate the three-dimensional positions of the body segments. However, this tracking approach needs controlled environments and limits the area of operation in a laboratory room, where the camera and the equipment are located [8].

As an alternative solution, the application of the accelerometer provides a simple means for analyzing gait in real-time for monitoring physical activity levels and classifying movements [9,10,11,12]. Gyroscopes were applied in the measurement of angular rate and the angle of various joints by mounting multiple devices on the lower or upper extremities [12,13,14,15]. In this paper, a wireless IMU sensor network (i.e., two IMUs) is used to enable a three-dimensional angular assessment of the thigh and the knee for gait analysis [16]. The differential changes of the rotation angles are also used to reduce the influences of the drift and offset noises that are common in IMU raw data. In particular, this paper presents gait phase differences using the sagittal, frontal, and transverse directional angles of the knee joint as a novel parameter to classify normal and abnormal gait patterns. The preliminary experimental results demonstrate that the proposed parameter is promising and can carry out gait pattern identification.

## 2. Wearable Inertial Sensor Network

### 2.1. Sensor System

This paper describes the method to acquire and analyze the IMU data so that an abnormal gait from a combination of hip and knee motion can be detected. The main system configuration is a pair of IMU sensors to create a network of joints and links of the human leg (Figure 1). The study uses Bluetooth technology for wireless transmission, and an Android device or a PC acts as a host receiver for the data transmitted using these sensors. They have a host application installed to allow the collection of data in the receiver. A MATLAB program involving mathematical calculations provides the analysis of the sensor data. 

The specifications of the wireless IMU system are as follows: The angular motion data from the IMU is in the form of quaternions.The wireless IMU sensor is small in size with a diameter of 23 mm.The wireless communication unit provides a low power communication interface between the data acquisition system and PC or any host where Bluetooth 4.0 is supported.A Nordic Semiconductor nRF51822 chip is used as a communication module.The nRF51822 communication system on chip (SoC) supports three kinds of data rates—2 Mbps, 1 Mbps, and 250 kbps.The transmission of raw signals has a sampling rate of 250 data points per seconds.The reconfigurable system interfaces the components using API (application programming interfaces) that can interface with 3rd party applications.An onboard Li-Polymer charge management controller MCP73831 recharges the sensor module.With Bluetooth Low Energy (BLE), two devices communicate only at each connection interval, so the minimum time between transfers will be the connection interval (between 7.5 ms to 4.0 s).The NRF24L01+ is a 2.4 GHz Radio-Frequency (RF) transceiver integrates an RF transceiver, a full-speed USB 2.0 compliant device controller, an 8-bit microcontroller, and flash memory.

The motion sensor module consists of a 9-axis sensor made by ‘Invensense’, MPU9150. The motion processor acquires the data collected by the accelerometer, gyroscope, and magnetometer, to generate the data in the format of quaternions. The MPU-9150 features three 16-bit analog-to-digital converters (ADCs) for digitizing the gyroscope with a full-scale range of ± 250, ± 500, ± 1000, and ± 2000°/s (dps). The noise rate is 0.005 (◦/s/√ Hz). However, our test showed about 5°/h stability of the sensor assembly at the 20 Hz sampling rate.

### 2.2. Gait Cycle

The body segments are referenced to the anatomical position and are described as occurring in three planes (Figure 2a). The body is divided into left and right halves by the sagittal plane. The frontal plane divides the body into anterior (front) and posterior (rear) portions. The body is divided into superior (upper) and inferior (lower) portions of the transverse plane. Joint motions during walking in the sagittal reference plane are shown in Figure 2b. The motion of the hip and knee in the sagittal plane are referred to as flexion and extension. Ankle movement in the sagittal plane is referred to as dorsiflexion and plantar flexion. Motions in the frontal plane are abduction and adduction, and motions in the transverse plane are internal and external rotations. Translating the center of mass of the human body is the primary task of walking. The sagittal plane is the parallel plane to the progression of walking. The left and right step during walking is a single gait cycle, which is a pattern of repeatable movements.

One gait cycle of human walking consists of two consecutive steps. A step is a period from right heel contact (RHC) to left heel contact (LHC). Two symmetric steps define a typical gait cycle containing single support (SS) and double support (DS) phase. During walking, the leg moves in a repeatable motion giving each person a unique walk. The leg moves by the multiple degrees of angular movements of the hip and knee joints to achieve a periodic pattern that enables the forward movement of the body [18,19]. Figure 2b shows that there is a time difference between the two peak swing motions of the hip and knee in the sagittal direction. It is assumed that phase differences exist in all the three-dimensional reference planes, which can characterize a unique gait cycle pattern. 

### 2.3. 3-Dimensional Hip and Knee Joint Angles

As shown in Figure 3, a human leg, as a kinematic manipulator, consists of two physical links (i.e., thigh and shin) and joints (i.e., hip and knee), each having three rotational joints. Both the physical joints are imaginary. Physical joint A is located at the hip (hip joint), and physical joint B is situated in the knee. Each physical joint has three Degree of Freedom (DOF) imparting a robot system. Links A and B are also imaginary and connect physical joint A to link A and physical Joint B to link B, respectively. Sensor A is placed at the end of Link A and gives the orientation of link A to the base frame. Sensor B is situated at the end of link B and provides the orientation of link B to the base frame. It should be noted that the direction of both the sensors is maintained the same.

By defining the transformation matrices $T_{A}^{0}$ as the mapping of sensor A in the base frame 0 and $T_{B}^{A}$ as the mapping of sensor B in the sensor A frame, they can be described as follows based on the Denavit and Hartenberg (D–H) convention. The four parameters of each link give a homogeneous transformation matrix $A_{i}$. As per D‒H convention law, every $A_{i}$ can be represented as a product of four basic transformations as shown below. 

(1)$$A_{i}=Rot_{z,\theta_{i}}\;Trans_{z,d_{i}}\;Trans_{x,a_{i}}\;Rot_{x,a_{i}}$$

(2)$$A_{i}=\left[\begin{array}{cccc} c_{\theta_{i}} & -s_{\theta_{i}} & 0 & 0\\ s_{\theta_{i}} & c_{\theta_{i}} & 0 & 0\\ 0 & 0 & 1 & 0\\ 0 & 0 & 0 & 1 \end{array}\right]\left[\begin{array}{cccc} 1 & 0 & 0 & 0\\ 0 & 1 & 0 & 0\\ 0 & 0 & 1 & d_{i}\\ 0 & 0 & 0 & 1 \end{array}\right]\left[\begin{array}{cccc} 1 & 0 & 0 & \alpha_{i}\\ 0 & 1 & 0 & 0\\ 0 & 0 & 1 & 0\\ 0 & 0 & 0 & 1 \end{array}\right]\left[\begin{array}{cccc} 1 & 0 & 0 & 0\\ 0 & c_{\alpha_{i}} & -s_{\alpha_{i}} & 0\\ 0 & s_{\alpha_{i}} & c_{\alpha_{i}} & 0\\ 0 & 0 & 0 & 1 \end{array}\right]$$

(3)$$A_{i}=\left[\begin{array}{cccc} c_{\theta_{i}} & -s_{\theta_{i}}c_{\alpha_{i}} & s_{\theta_{i}}s_{\alpha_{i}} & a_{i}c_{\theta_{i}}\\ s_{\theta_{i}} & c_{\theta_{i}}c_{\alpha_{i}} & -c_{\theta_{i}}s_{\alpha_{i}} & a_{i}s_{\theta_{i}}\\ 0 & s_{\alpha_{i}} & c_{\alpha_{i}} & d_{i}\\ 0 & 0 & 0 & 1 \end{array}\right]$$

Here, ‘s’ stands for sine and ‘c’ stands for cosine of the respective angles. Equation (3) is the general form of the homogeneous transformation matrix, which varies with each link, based on the D‒H parameters of that link. 

The transformation from frame 0 to frame 1 of the model can now be obtained by simply substituting the D–H parameters of link 1 in the general form of the homogeneous transformation matrix. Likewise, the homogeneous transformation matrix can be derived for link 1 through 6. Let $A_{1}^{0}$ be the transformation from frame 0 to frame 1 or the mapping of frame 1 in frame 0. Then,

(4)$$A_{1}^{0}=\left[\begin{array}{cccc} c_{\theta_{1}} & 0 & -s_{\theta_{1}} & 0\\ s_{\theta_{1}} & 0 & c_{\theta_{1}} & 0\\ 0 & -1 & 0 & 0\\ 0 & 0 & 0 & 1 \end{array}\right].$$

Similarly, 

(5)$$A_{2}^{1}=\left[\begin{array}{cccc} c_{\theta_{2}} & 0 & s_{\theta_{2}} & 0\\ s_{\theta_{2}} & 0 & -c_{\theta_{2}} & 0\\ 0 & 1 & 0 & 0\\ 0 & 0 & 0 & 1 \end{array}\right]\;,\;A_{3}^{2}=\left[\begin{array}{cccc} c_{\theta_{3}} & -s_{\theta_{3}} & 0 & 0\\ s_{\theta_{3}} & c_{\theta_{3}} & 0 & 0\\ 0 & 0 & 1 & d_{3}\\ 0 & 0 & 0 & 1 \end{array}\right]$$

(6)$$A_{4}^{3}=\left[\begin{array}{cccc} c_{\theta_{4}} & 0 & -s_{\theta_{4}} & 0\\ s_{\theta_{4}} & 0 & c_{\theta_{4}} & 0\\ 0 & -1 & 0 & 0\\ 0 & 0 & 0 & 1 \end{array}\right],\;A_{5}^{4}=\left[\begin{array}{cccc} c_{\theta_{5}} & 0 & s_{\theta_{5}} & 0\\ s_{\theta_{5}} & 0 & -c_{\theta_{5}} & 0\\ 0 & 1 & 0 & 0\\ 0 & 0 & 0 & 1 \end{array}\right],$$

(7)$$A_{6}^{5}=\left[\begin{array}{cccc} c_{\theta_{6}} & -s_{\theta_{6}} & 0 & 0\\ s_{\theta_{6}} & c_{\theta_{6}} & 0 & 0\\ 0 & 0 & 1 & d_{6}\\ 0 & 0 & 0 & 1 \end{array}\right].$$

Since there are only 2 physical joints, A and B, each having 3 rotational joints whose origins coincide, the number of homogeneous transformation matrices can also be reduced from 6 to only 2. The mapping of the 1st physical joint to frame 0 is $T_{3}^{0}$. Let $T_{3}^{0}$ be the transformation from frame 0 to frame 3 or, analogously, the mapping of frame 3 in frame 0. Since the model distinctly consists of two physical links and joints, each having three rotational joints, the number of transformation matrices can be reduced from 6 to 2 as follows: (8)$$T_{A}^{0}=A_{1}^{0}\;A_{2}^{1}\;A_{3}^{2},$$
(9)$$T_{A}^{0}=\left[\begin{array}{cccc} c_{1}c_{2}c_{3}-s_{1}s_{3} & -c_{1}c_{2}s_{3}-s_{1}c_{3} & c_{1}s_{2} & d_{A}(c_{1}s_{2})\\ s_{1}c_{2}c_{3}+c_{1}s_{3} & -s_{1}c_{2}s_{3}+c_{1}c_{3} & s_{1}s_{2} & d_{A}(s_{1}s_{2})\\ -s_{2}c_{3} & s_{2}s_{3} & c_{2} & d_{A}(c_{2})\\ 0 & 0 & 0 & 1 \end{array}\right],$$
(10)$$T_{B}^{A}=A_{4}^{3}\;A_{5}^{4}\;A_{6}^{5},$$
(11)$$T_{B}^{A}=\left[\begin{array}{cccc} c_{4}c_{5}c_{6}-s_{4}s_{6} & -c_{4}c_{5}s_{6}-s_{4}c_{6} & c_{4}s_{5} & d_{B}(c_{4}s_{5})\\ s_{4}c_{5}c_{6}+c_{4}s_{6} & -s_{4}c_{5}s_{6}+c_{4}c_{6} & s_{4}s_{5} & d_{B}(s_{4}s_{5})\\ -s_{5}c_{6} & s_{5}s_{6} & c_{5} & d_{B}(c_{5})\\ 0 & 0 & 0 & 1 \end{array}\right],$$
where $c_{1}$, $c_{2}$, $c_{3}$, $c_{4}$, $c_{5}$, $c_{6}$ stand for cos ($\theta_{1}$), cos ($\theta_{2}$), cos ($\theta_{3}$), cos ($\theta_{4}$), cos ($\theta_{5}$), cos ($\theta_{6}$), and $s_{1}$, $s_{2}$, $s_{3}$, $s_{4}$, $s_{5}$, $s_{6}$ are sin $\left(\theta_{1}\right)$, sin ($\theta_{2}$), sin ($\theta_{3}$), sin $\left(\theta_{4}\right)$, sin ($\theta_{5}$), sin ($\theta_{6}$), respectively. 

The global homogeneous transformation matrix $T_{0}^{G}$ is assumed to be an identity matrix,

(12)$$T_{0}^{G}=\left[\begin{array}{cccc} 1 & 0 & 0 & 0\\ 0 & 1 & 0 & 0\\ 0 & 0 & 1 & 0\\ 0 & 0 & 0 & 1 \end{array}\right].$$

The forward kinematic equation to obtain the sensor B orientation in/with respect to base frame (frame 0) is given as follows: (13)$$T_{B}^{G}=T_{0}^{G}\;*\;T_{A}^{0}\;*\;T_{B}^{A}.$$

Rearranging the above equation to get orientation of sensor B with respect to the sensors A [11] or equivalently the orientation of frame 6 with respect to frame 3:(14)$$T_{B}^{A}={\left(T_{A}^{0}\right)}^{-1}*{\left(T_{0}^{G}\right)}^{-1}*T_{B}^{0}$$
where $T_{B}^{0}$ is the initialized version of $T_{B\;\left(original\right)}^{0}$. 

The initialization of rotation matrix obtained from the sensors A and B is done in order to perfectly align sensor B ($T_{B}^{0}$) with sensor A ($T_{A}^{0}$) in the sagittal plane. To initialize $T_{A\;\left(original\right)}^{0}$ and $T_{B\;\left(original\right)}^{0}$ in the mid-sagittal plane, the rotation matrices of sensors A and B at the initially aligned position of the leg in the sagittal plane are calculated. Thus, the calibrated rotation matrix of sensor A ($T_{A}^{0}$) is given by

(15)$$T_{A\;\left(inv\right)}^{0}={\left[T_{A\;\left(original\right)}^{0}\left(at\;initial\;position\right)\right]}^{-1},$$

(16)$$T_{A}^{0}=T_{A\;\left(original\right)}^{0}*T_{A\;\left(inv\right)}^{0}.$$

Similarly, the calibrated rotation matrix of sensor B ($T_{B}^{0}$) is given by

(17)$$T_{B\;\left(inv\right)}^{0}={\left[T_{B\;\left(original\right)}^{0}\left(at\;initial\;position\right)\right]}^{-1},$$

(18)$$T_{B}^{0}=T_{B\;\left(original\right)}^{0}*T_{B\;\left(inv\right)}^{0}.$$

Note that, in Equations (16) and (18), only the inverse of the first matrices of $T_{A\;\left(original\right)}^{0}$ and $T_{B\;\left(original\right)}^{0}$ are taken so that the first matrices of initialized $T_{A}^{0}$ and $T_{B}^{0}$ become an identity matrix always.

Next, by applying inverse kinematics to $T_{A}^{0}$, the 3-D hip joint angles $\theta_{1}$, $\theta_{2}$, $\theta_{3}$ are obtained.

(19)$$r_{33}=c_{2},\;\;\theta_{2}=\cos^{-1}(r_{33})$$

(20)$$r_{31}=-s_{2}c_{3},\;\;\theta_{3}=\cos^{-1}\left(\frac{-r_{31}}{s_{2}}\right)$$

(21)$$r_{13}=c_{1}s_{2},\;\;\theta_{1}=\cos^{-1}\left(\frac{r_{13}}{s_{2}}\right)$$

Similarly, by applying inverse kinematics to $T_{B}^{A}$, the 3-D knee joint angles $\theta_{4}$, $\theta_{5}$, $\theta_{6}$ (Figure 3b) can by calculated by 

(22)$$r_{33}=c_{5},\;\;\theta_{5}=\cos^{-1}(r_{33}),$$

(23)$$r_{31}=-s_{5}c_{6},\;\;\theta_{6}=\cos^{-1}\left(\frac{-r_{31}}{s_{5}}\right),$$

(24)$$r_{13}=c_{4}s_{5},\;\;\theta_{4}=\cos^{-1}\left(\frac{r_{13}}{s_{5}}\right).$$

## 3. Experimental Results

Normal gait requires sophisticated interaction of many systems of the body, such as strength, coordination, and sensation. For example, aging is an increasing global public health issue. When aging causes weakening strength and sensation, the interacting systems may result in abnormal gait or walking abnormality. In this section, a preliminary experimental result for the recognition of normal and abnormal gait types from the calculated sets of 3-D hip and knee joint angles (Equations (12)–(17)) are described. Hemiplegic gait is one of the results of post-stroke patients who have weak flexor muscle movement during the gait cycle. The knee is stiff, hyperextends during stance, and does not regularly flex during the swing. The contralateral step often advances to meet the position of the paralyzed limb, instead of periodically moving forward beyond it. These movements include raising the pelvis to clear the paralyzed leg and circumduction, an abnormal shift. As a result, the toe traces a semicircle on the floor, first moving outward and then inward as it advances, instead of a normal straightforward movement [20,21,22,23] (Figure 4).

This experiment provides the results of three-dimensional gait analysis performed on a single participant (i.e., an author). The experiments were performed multiple times to ensure the repeatability of the results and to obtain a statistically significant data set. A PC was then used to record and process the data. The experiments were completed the same way each time but on different days. There was only a single individual that performed the test, and thus, the sampling size was not statistically significant. However, for proof of concept purposes, the test was successful and serves as motivation to repeat the tests on a larger sample size in the future. Figure 5 shows the two kinds of gait patterns that were considered: hemiplegic and normal gaits. Three-dimensional kinematics of the hip and knee joint angles were recorded and calculated. The figure shows the relative angles of the hip and knee during the gait cycle along with extension and flexion in the sagittal plane, the frontal plane, and the transverse plane. From the figure, it can be noticed the changed coordination phase of the hip and knee movements of the Hemiplegic gait.

In this paper, a new method to calculate the parameter of the three-dimensional phase difference angles between the hip and knee movements is described to identify biomechanical changes in gait patterns. The phase difference angle analysis aims to establish the distinct conditions for mapping of the characteristic patterns of the joint motions in the performing of specific leg movements. The phase difference angles at time t and the *i*th-directional plane (i.e., sagittal, frontal, and transverse) is given by
(25)$$\varphi_{t}^{i}=360{}^{\circ}\;f\;\Delta t,$$
where *f* is gait cycle frequency and Δt is the time difference (PD) between the hip and the knee angles.

This study uses the instantaneous signal (i.e., the temporal derivative of the oscillation) since it is useful for describing the non-monochromatic nature of the joint angle signals. Further, it also helps to reduce the effect on noises in IMU signal measurement, such as the signal shift and offset errors. Figure 6 shows the normalized gait phase diagrams of the hip and knee joint angles of Figure 5. From the diagram, the phase deference angles (i.e., hip-sagittal vs. knee-sagittal; hip-sagittal vs. knee-frontal; hip-sagittal vs. knee-transverse) are calculated using Equation (25). All phase deference angles of the two gait patterns are summarized in Figure 7. The figure indicates that all the sagittal, frontal, and transverse directions show a different pattern between the normal and the hemiplegic gait of the subject. Table 1 summarizes the experimental results from multiple trials of the subject.

## 4. Discussion

The gait experiment with the two-IMU network system showed promising results in measuring three-dimensional joint angles. One of the main contributions of this paper is to provide a procedure to calculate the three-dimensional relative angles of the knee joint with respect to the hip joint using a pair of IMUs. It seems that the IMU-based method has lower accuracy compared to an optical camera-based method, such as the Vicon system [24]. However, the preliminary experimental results showed similar waveforms of hip and knee angles with comparable peak heights [24]. Unknown initial joint angles and error accumulation in the integral value of the gyroscope were still a significant problem. Further, there were some minor differences when observing the results, such as some abrupt changes at peaks or valleys during motion. There could be several reasons for the differences: human error or sensitivity of the sensors. Human error could be from mounting the sensors. If there was some error from the mounting, it did not appear to be significant.

In the identification of the characteristics of human gait, the study proposes using phase difference angle patterns. Figure 7 and Table 1 show evidence of the phase difference angle changes between the hip and joint movements during hemiplegic walking. Table 1 shows that the phase difference angles in all the directions were decreasing consistently.

It is known that a stroke commonly results in hemiplegic gait cycle impairments that are associated with decreased hip and knee coordination and limited leg muscle strength. These impairments often result in biomechanical changes during walking. The figures reflect the effects of reduced control of the leg swing that results in the decreased phase difference angles. The three-dimensional phase difference angle patterns are suitable for training machine learning methods to detect gait changes during walking [25,26].

## 5. Conclusions

The work contributes to classifying the characteristic joint movement patterns of the hip and the knee joints during the walking. The study conducted the following three steps to recognize the phase difference angle changes: i) calculation of 3-D joint angles of the hip and knee from the two-IMU sensor network system; ii) calculation of the normalized gait phase diagrams; iii) estimation of the gait phase difference angles for classifying abnormality in gait patterns. The work contributes to IMU signal analysis by introducing the following techniques: instantaneous signal normalization to reduce noise from the acquisition and the gait phase difference angle as a technique for the decomposition of signals. The methods proposed in the paper are suitable for establishing the patterns of different gait movements. The proposed techniques contribute in many ways to state-of-the-art identification, modeling, decomposition, and prediction of IMU signal analysis for various body motion applications. 

## Figures and Tables

**Figure 1 sensors-19-05024-f001:**
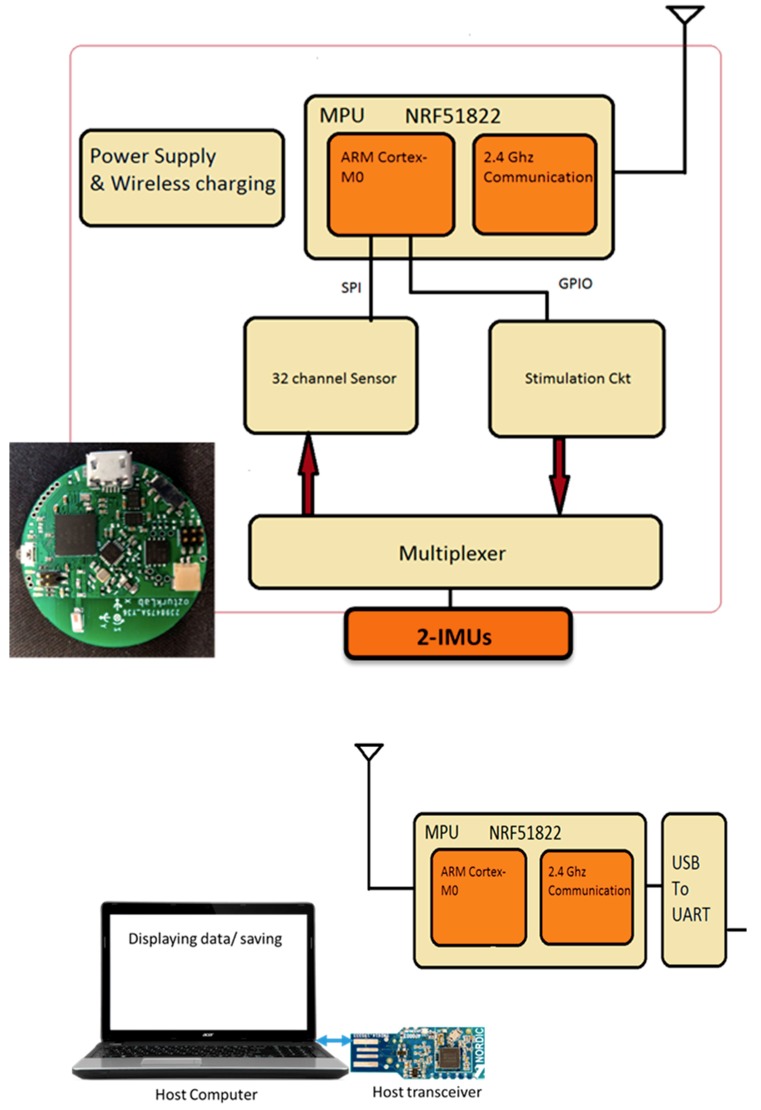
A custom designed IMU network to record leg movements: the sensor system diagram and the manufactured wireless IMU sensor circuit.

**Figure 2 sensors-19-05024-f002:**
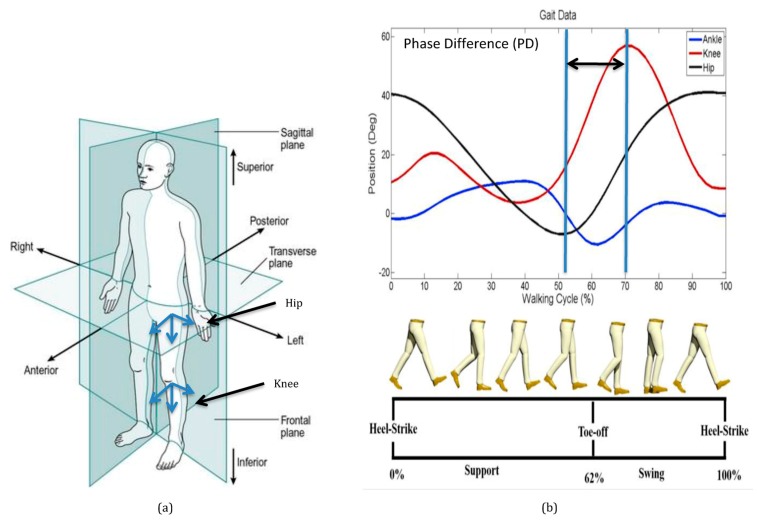
One gait cycle of human walking: (**a**) description of the hip and knee local coordinate system that are referenced to the anatomical position in three reference planes; (**b**) description of the phase difference motion of the hip and knee in the sagittal plane [17].

**Figure 3 sensors-19-05024-f003:**
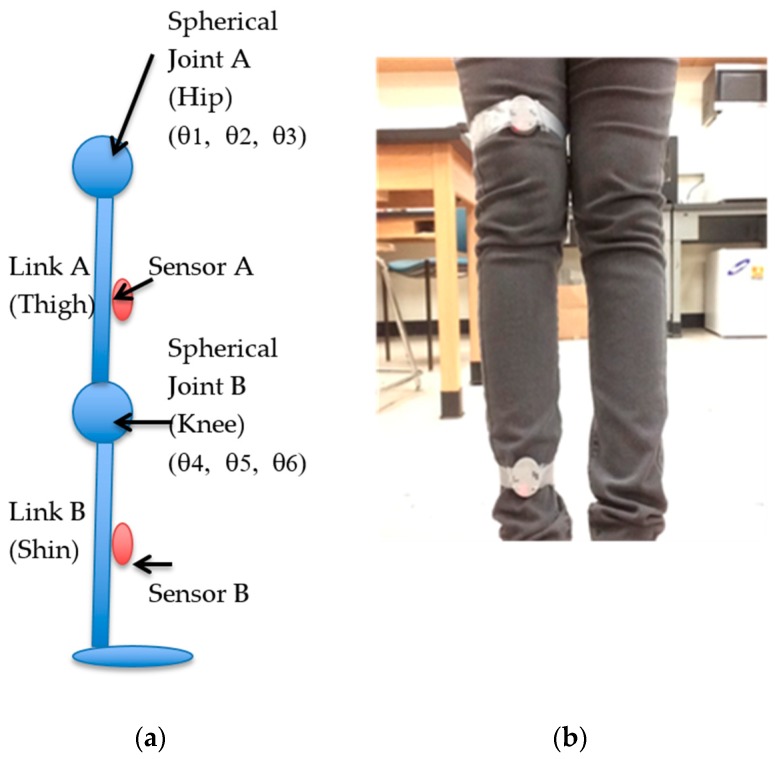
Three-dimensional joint angles: (**a**) description of a human leg model as a kinematic manipulator that consists of two physical links (i.e., thigh and shin) and joints (i.e., hip and knee), each having three rotational joints, the transformation; (**b**) an experimental sensor network setup.

**Figure 4 sensors-19-05024-f004:**
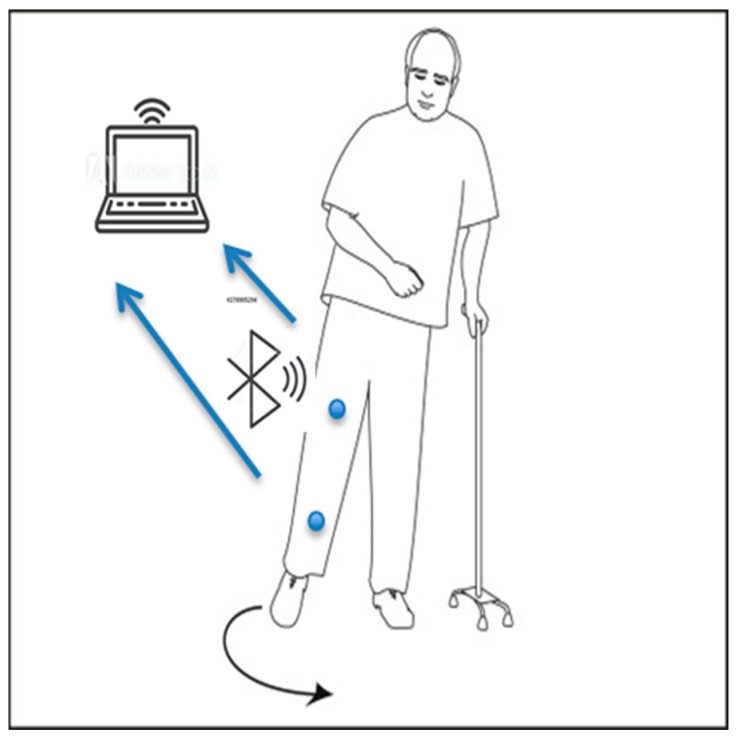
Description of a human leg movement during hemiplegic gait experiment.

**Figure 5 sensors-19-05024-f005:**
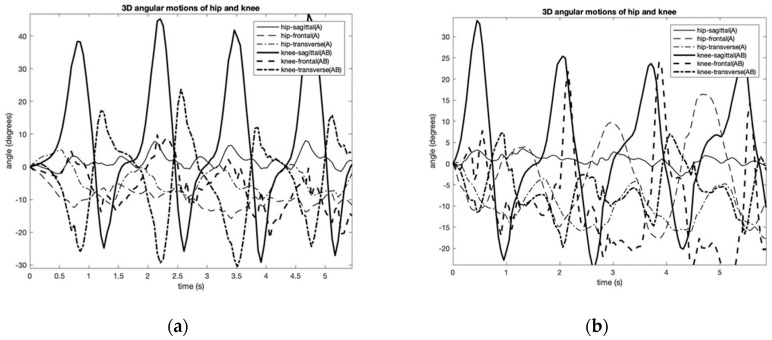
An example of calculated 3-D hip and knee joint angle patterns associated with normal (**a**) (periods of 1.3 s) and simulated hemiplegic walking (**b**) (periods of 1.8 s).

**Figure 6 sensors-19-05024-f006:**
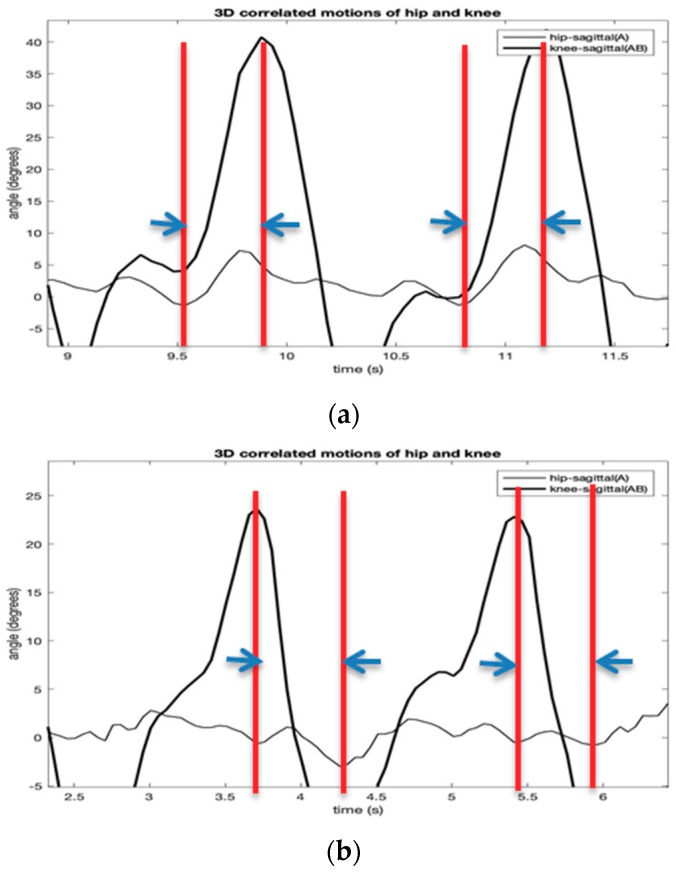
The normalized gait phase diagrams of the hip and knee joint angles of Figure 5 with normal (**a**) and simulated hemiplegic walking (**b**) (hip-sagittal vs. knee-sagittal).

**Figure 7 sensors-19-05024-f007:**
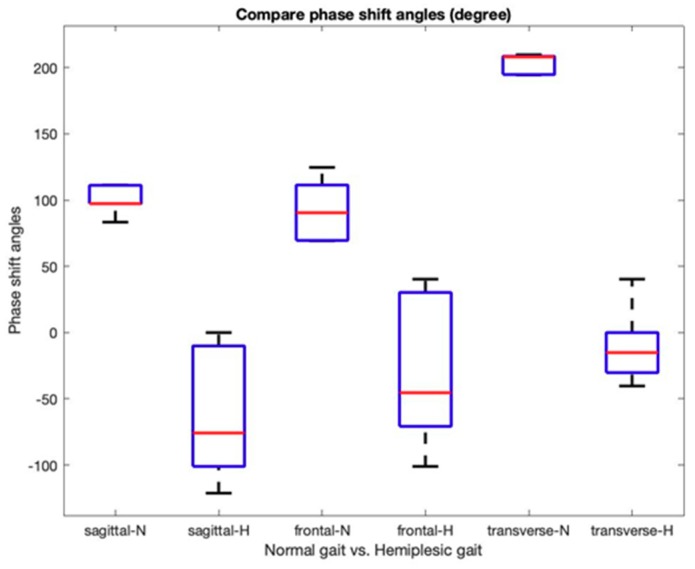
The normalized gait phase diagrams of the hip and knee joint angles of Figure 5 (i.e., hip-sagittal vs. knee-sagittal).

**Table 1 sensors-19-05024-t001:** The experimental results of the normalized gait phase angles with normal and simulated hemiplegic walking.

	Gait	Hip-Sagittal VS. Knee-Sagittal (deg.)	Hip-Sagittal VS. Knee-Frontal (deg.)	Hip-Sagittal VS. Knee-Transverse (deg.)
Normal	AVG ^1^	101.41	91.70	204.37
Normal	STD ^1^	9.39	21.90	6.69
Hemiplegic	AVG ^2^	−62.61	−27.22	−10.26
Hemiplegic	STD ^2^	45.62	54.20	7.04

^1^ The gait cycle frequency is 0.77 Hz. ^2^ The gait cycle frequency is 0.56 Hz.

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
