# Peer review of "Identification of Gait Motion Patterns Using Wearable Inertial Sensor Network"

_sensors, 2019, doi:10.3390/s19225024_

Round 1

Reviewer 1 Report

The authors developed a gait sensor network system using a pair of wireless IMU sensors to analyze the 3D angular motions of hip, knee, and ankle joints.
Although importance of the wearable motion analysis technology is well-known, there are several major concerns in this study, especially for system validation.

1) Introduction
- Recent advances of the wearable motion analysis technology should be reviewed more although two references [1, 2] were mentioned.
- Since many wearable systems have been developed. The sentence,  "However, human gait identification outside of the lab is still a difficult task," need to be clarified with references, and the novelty of this study should be described.

2) Wearable Inertial Sensor Network
- In 2.1, the detailed information for the sensor system, such as specification of the IMU sensor, needs to be provided.
- In 2.3, Some terms in the formulations were not explained, e.g., d_A in Eqs.(1),(2) and R_3*3 in Eqs.(4),(5). In addition, derivation of Eq.(7) from Eq.(6) is unclear.
- In Figure 3(a), the coordinate system direction was not common. Z-direcion should be reversed.

3) Experimental Results
- It is the major concern that validation procedure of the developed system was skipped, which is an essential part to present a new sensor system. 

Author Response

We are grateful to the reviewers for their insightful comments on our paper. We have been able to incorporate changes to reflect most of the suggestions provided by the reviewers. We have highlighted the changes in the red color within the manuscript. Please find a point-by-point response to your concerns. We hope that you find our responses satisfactory and that the paper is now acceptable for publication.

Comments and Suggestions for Authors

The authors developed a gait sensor network system using a pair of wireless IMU sensors to analyze the 3D angular motions of hip, knee, and ankle joints.

Although importance of the wearable motion analysis technology is well-known, there are several major concerns in this study, especially for system validation.

1) Introduction

- Recent advances of the wearable motion analysis technology should be reviewed more although two references [1, 2] were mentioned.

(Thank you for pointing this out. We agree with this comment. Therefore, we have added seven more references in the revised manuscript. This change can be found in line numbers 62-71 and references [9-15.])

- Since many wearable systems have been developed. The sentence, "However, human gait identification outside of the lab is still a difficult task," need to be clarified with references, and the novelty of this study should be described.

(Agree. We have accordingly removed the sentence.)

2) Wearable Inertial Sensor Network

- In 2.1, the detailed information for the sensor system, such as specification of the IMU sensor, needs to be provided.

(Thank you for pointing this out. We agree with this comment. Therefore, we have accordingly revised the paragraph. This change can be found line numbers 82-99.)

- In 2.3, Some terms in the formulations were not explained, e.g., d_A in Eqs.(1),(2) and R_3*3 in Eqs.(4),(5). In addition, derivation of Eq.(7) from Eq.(6) is unclear.

(Agree. We have accordingly revised the section. This change can be found line numbers 171-190.)

- In Figure 3(a), the coordinate system direction was not common. Z-direcion should be reversed.

(Agree. We have accordingly revised the figure.)

3) Experimental Results

- It is the major concern that validation procedure of the developed system was skipped, which is an essential part to present a new sensor system.

(Thank you for pointing this out. We agree with this comment. Therefore, we have added Table 1 and the new discussion section in the revised manuscript. This change can be found in line numbers 353-359 and 376-399.)

Reviewer 2 Report

The paper contains lengthy discussions on basics and the authors work is decried very short. Especially, the quality of the write-up in the latter part of the paper is poor. 

The order of references is not as per the standard. Reverences 3-4 (line 45 and 48) are not in correct order. "We/us/our" are used In many places (lines 59,67, 96, 103, 188, 206, 225, 255, 267). Recommended to use passive voice. Figures are not clear. Figures are addressed in some places as "Fig" and some places as "Figure". It is recommended to be consistent and use either of the methods. Text continuing after an equation should start with a simple letter (lines 150, 184, 265). Further, text should be left aligned in such cases. Fig 2 is taken from a reference without citing. Please cite the original source. The basis for the content in section 2.3 is not clearly mentioned. Is it a derivation from previous literature or a duplicate. Line 178: What is "Eq. XX"? Line: 184: T60 from where has it come? Line 222: delete "are" Figure 7: what is given in the horizontal axis. Details of gait phase diagram is not clear. Please clearly describe what you have depicted. Line 335 mentions a camera system. However, there is no mention of a reference system and how the reference system is used to validate the experimental results. Please explain clearly. Line 346: Delete "Please add". Line 349: "their" or "his/her"?

The experiment is conducted using only 1 subject, which I think is not sufficient. Even for the single subject, the number of trials, stride cycles for each trial and where the trials were recorded is not clearly mentioned. As such, it is difficult to judge if the data set is statistically sound.

It is recommended to conduct the experiments using several subjects and include statistical data of the accuracy of results before the paper can be accepted.

Significant improvements are necessary to accept this paper for publication in a high ranked journal like Sensors.

Author Response

We are grateful to the reviewers for their insightful comments on our paper. We have been able to incorporate changes to reflect most of the suggestions provided by the reviewers. We have highlighted the changes in the red color within the manuscript. Please find a point-by-point response to your concerns. We hope that you find our responses satisfactory and that the paper is now acceptable for publication.

Comments and Suggestions for Authors

The paper contains lengthy discussions on basics and the authors work is decried very short. Especially, the quality of the write-up in the latter part of the paper is poor.

The order of references is not as per the standard. Reverences 3-4 (line 45 and 48) are not in correct order. "We/us/our" are used In many places (lines 59,67, 96, 103, 188, 206, 225, 255, 267). Recommended to use passive voice. Figures are not clear. Figures are addressed in some places as "Fig" and some places as "Figure". It is recommended to be consistent and use either of the methods. Text continuing after an equation should start with a simple letter (lines 150, 184, 265). Further, text should be left aligned in such cases. Fig 2 is taken from a reference without citing. Please cite the original source. The basis for the content in section 2.3 is not clearly mentioned. Is it a derivation from previous literature or a duplicate. Line 178: What is "Eq. XX"? Line: 184: T60 from where has it come? Line 222: delete "are" Figure 7: what is given in the horizontal axis. Details of gait phase diagram is not clear. Please clearly describe what you have depicted. Line 335 mentions a camera system. However, there is no mention of a reference system and how the reference system is used to validate the experimental results. Please explain clearly. Line 346: Delete "Please add". Line 349: "their" or "his/her"?

The experiment is conducted using only 1 subject, which I think is not sufficient. Even for the single subject, the number of trials, stride cycles for each trial and where the trials were recorded is not clearly mentioned. As such, it is difficult to judge if the data set is statistically sound.

(Thank you for pointing this out. We agree with this comment. Therefore, we have accordingly revised the manuscript. This change can be found in the heightened in red colors.)

It is recommended to conduct the experiments using several subjects and include statistical data of the accuracy of results before the paper can be accepted.

(Thank you for pointing this out. We agree with this comment. Therefore, we have added Table 1 and the new discussion section in the revised manuscript. This change can be found in line numbers 353-359 and 376-399. Please understand that the IRB required including a single subject for this study.)

Significant improvements are necessary to accept this paper for publication in a high ranked journal like Sensors.

Reviewer 3 Report

The study by Kee and colleagues investigated Gait motion patterns using wearable IMU sensor. First of all, this study is methodologically sound. But, need to improve the previous related works in the Introduction part. Also, the implications and relevance of the work are currently concealed due to a poor framing of the problem space and very limited discussion of the findings and their relevance. Therefore, the introduction (rationale) and discussion (interpretation) need to be strengthen to highlight the significance of the present investigation. Eventually, there is no Discussion part in the manuscript.

Some specific comments putted below:

The Abstract should state the rationale for the study, main objective, methods used, summary of findings (recommended numeric values), and brief conclusions in a paragraph. The rationale needs to be more convincing.  As it stands, the main thrust of the introduction is that previous research in this area is limited, which is not a convincing reason for why it should be conducted now.  I suggest you clearly explain the benefit to the practitioner, and show how this study can be developed and incorporated to inform, evaluate, and design training. Figure 6 need to redraw, very unclear and hard to understand. Were the data normally distributed? Was there a check for this? Has this IMU device been validated before against lab standards The sentence closing with "etc." is unfathomable. A discussion (one paragraph) of the implications of this work for clinicians/ researchers is required.

Author Response

We are grateful to the reviewers for their insightful comments on our paper. We have been able to incorporate changes to reflect most of the suggestions provided by the reviewers. We have highlighted the changes in the red color within the manuscript. Please find a point-by-point response to your concerns. We hope that you find our responses satisfactory and that the paper is now acceptable for publication.

Comments and Suggestions for Authors

The study by Kee and colleagues investigated Gait motion patterns using wearable IMU sensor. First of all, this study is methodologically sound. But, need to improve the previous related works in the Introduction part. Also, the implications and relevance of the work are currently concealed due to a poor framing of the problem space and very limited discussion of the findings and their relevance. Therefore, the introduction (rationale) and discussion (interpretation) need to be strengthen to highlight the significance of the present investigation. Eventually, there is no Discussion part in the manuscript.

(Thank you for pointing this out. We agree with this comment. Therefore, we have revised the introduction and the new discussion section in the revised manuscript. This change can be found in line numbers 62-71 and 376-399.)

Some specific comments putted below:

The Abstract should state the rationale for the study, main objective, methods used, summary of findings (recommended numeric values), and brief conclusions in a paragraph. The rationale needs to be more convincing. As it stands, the main thrust of the introduction is that previous research in this area is limited, which is not a convincing reason for why it should be conducted now. I suggest you clearly explain the benefit to the practitioner, and show how this study can be developed and incorporated to inform, evaluate, and design training. Figure 6 need to redraw, very unclear and hard to understand. Were the data normally distributed? Was there a check for this? Has this IMU device been validated before against lab standards The sentence closing with "etc." is unfathomable. A discussion (one paragraph) of the implications of this work for clinicians/ researchers is required.

(Thank you for pointing this out. We agree with this comment. Therefore, we have added Table 1 and the new discussion section in the revised manuscript. This change can be found in line numbers 353-359 and 376-399. Please understand that the IRB required including a single subject for this study. Also, we have revised accordingly the abstract and the figures in the revised manuscript. This change can be found in line numbers 13-26, and 301-344.)

Round 2

Reviewer 1 Report

The authors revised the manuscript well according to the reviewer.
However, there are still a few concerns as follows:

1) Introduction
- The novelty of this study is weak though the last sentence in the Introduction mentioned as "In particular, this paper presents gait phase differences using the sagittal, frontal, and transverse directional angles of the knee joint as a novel parameter to classify normal and abnormal gait patterns." A short sentence may be necessary to provide the advantage of the novel parameter in comparison to previous parameters.

2) Wearable Inertial Sensor Network
- In 2.1, the revised paragraph was a rearrangement of previous version.
The resolution of IMU system that explain such as the accuracy of measurement needs to be presented.

- In 2.3, some parts were still not clarified. For example, A^0_1, A^1_2, A^2_3, in Eq.(4) and d_A in Eq.(5) were not explained. In addition, derivation of Eq.(10) from Eq.(9) is still unclear. In the sentence after Eq.(10), "Where, T^0_6 is the initialized version of T^0_6(original)," T^0_6 was not previously used. In Figure 3(a), the coordinate system direction was not properly corrected (Z-direcion should be the same to the cross-product of X- and Y- directions. If not, the signs of phi_3 and phi_6 may be reversed in the calculation.) In addition, the Figure 3(a) needs to be carefully revised. Some characters in the previous version, e.g. d_3, d_6, x, y, and z, were still shown.

3) Experimental Results
- Although the Table 1 showed all data from three trials, it did not guarantee the validity of the measurement using the developed system. The authors mentioned in the Discussion, "The three-dimensional gait analysis test results came close to the results derived from using camera systems."
However, there was no direct information regarding the results derived from using camera systems. At least, the validation of the developed system should be discussed by comparing the results in this study to those
from using camera systems of previous studies.

Author Response

We are grateful for your constructive comments on our paper. We have been able to incorporate changes to reflect your suggestions. We have highlighted the changes in the red color within the manuscript. Please also find a point-by-point response to your concerns. We hope that you find our responses satisfactory and that the paper is now acceptable for publication.

Comments and Suggestions for Authors

1) Introduction

The novelty of this study is weak though the last sentence in the Introduction mentioned as "In particular, this paper presents gait phase differences using the sagittal, frontal, and transverse directional angles of the knee joint as a novel parameter to classify normal and abnormal gait patterns." A short sentence may be necessary to provide the advantage of the novel parameter in comparison to previous parameters.

(Thank you for pointing this out. We understand your suggestion. Therefore, we have added a sentence in the revised manuscript. This change can be found in line numbers 71-72.)

2) Wearable Inertial Sensor Network

- In 2.1, the revised paragraph was a rearrangement of previous version.

The resolution of IMU system that explain such as the accuracy of measurement needs to be presented.

(Thank you for pointing this out. We agree with this comment. Therefore, we have added a paragraph. This change can be found in line numbers 133-138.

In 2.3, some parts were still not clarified. For example, A^0_1, A^1_2, A^2_3, in Eq.(4) and d_A in Eq.(5) were not explained. In addition, derivation of Eq.(10) from Eq.(9) is still unclear. In the sentence after Eq.(10), "Where, T^0_6 is the initialized version of T^0_6(original)," T^0_6 was not previously used.

(Agree. Equations (4)-(7) were added to add the explanation. This change can be found line numbers 194-202 and 248.)

In Figure 3(a), the coordinate system direction was not properly corrected (Z-direcion should be the same to the cross-product of X- and Y- directions. If not, the signs of phi_3 and phi_6 may be reversed in the calculation.) In addition, the Figure 3(a) needs to be carefully revised. Some characters in the previous version, e.g. d_3, d_6, x, y, and z, were still shown.

(Agree. Figure 3(a) was replaced by a new figure. This change can be found in line numbers 215-240.)

3) Experimental Results

- Although the Table 1 showed all data from three trials, it did not guarantee the validity of the measurement using the developed system.

(Thank you for pointing this out. We agree with this comment. A new experiment was carefully designed and executed the test by a different person for validation. Please see the latest figures, including a statistical box plot (Figure 7). The new data set is shown in Table 1. This change can be found in line numbers 354-381.)

- The authors mentioned in the Discussion, "The three-dimensional gait analysis test results came close to the results derived from using camera systems."

However, there was no direct information regarding the results derived from using camera systems. At least, the validation of the developed system should be discussed by comparing the results in this study to those

from using camera systems of previous studies.

(The statement was revised. This change can be found in line numbers 394-399.)

Reviewer 2 Report

Authors have improved the paper addressing some of the points made in the 1st review. However, there are still areas that are not addressed.

Text after equations need to be corrected. Reference of Figure 2 is not cited in the caption. Horizontal axis of Figure 7 is not labeled Gait phase diagrams in Figure 6 are not clearly explained. Need more explanation.

Further, according to Table 1, this paper is based on 3 trails per each different walking style and all 3 trails are done by a single subject. No details are given of the exact sample size though. 3 samples cannot be considered as a statistically sound sample size. Although, according to the recommendations of the funding organization, only 1 subject could be used, the authors should have conducted many trails so as to make the data set statistically sound. Please include if details are available.

The authors have not clearly mention the accuracy of their results compared to the optical reference. They have just mentioned "The three-dimensional gait analysis test results came close to the results derived from using camera systems." which does not indicate the exact accuracy. It is recommended to use some statistical representation such as MSE to indicate the accuracy compared to the optical reference.

Author Response

We are grateful for your constructive comments on our paper. We have been able to incorporate changes to reflect your suggestions. We have highlighted the changes in the red color within the manuscript. Please also find a point-by-point response to your concerns. We hope that you find our responses satisfactory and that the paper is now acceptable for publication.

Comments and Suggestions for Authors

Text after equations need to be corrected. Reference of Figure 2 is not cited in the caption. Horizontal axis of Figure 7 is not labeled Gait phase diagrams in Figure 6 are not clearly explained. Need more explanation.

(Thank you for pointing this out. We agree with this comment. Figures 2,6, and 7 were replaced by new figures. This change can be found in line numbers 151-167, 330-353, and 363-381.)

Further, according to Table 1, this paper is based on 3 trails per each different walking style and all 3 trails are done by a single subject. No details are given of the exact sample size though. 3 samples cannot be considered as a statistically sound sample size. Although, according to the recommendations of the funding organization, only 1 subject could be used, the authors should have conducted many trails so as to make the data set statistically sound. Please include if details are available.

(Thank you for pointing this out. We agree with this comment. A new experiment was carefully designed and executed the test by a different person for validation. Please see the latest figures, including a statistical box plot (Figure 7). The new data set is shown in Table 1. This change can be found in line numbers 354-381.)

The authors have not clearly mention the accuracy of their results compared to the optical reference. They have just mentioned "The three-dimensional gait analysis test results came close to the results derived from using camera systems." which does not indicate the exact accuracy. It is recommended to use some statistical representation such as MSE to indicate the accuracy compared to the optical reference.

Thank you for pointing this out. We agree with this comment. Therefore, the statement was revised. This change can be found in line numbers 394-399.)

Reviewer 3 Report

I am happy with the changes did by the authors. I have no further comments.

Author Response

We are grateful for your constructive comments on our paper.

Round 3

Reviewer 1 Report

The authors revised the manuscript properly according to the reviewer's comments. So, the manuscipt is acceptable for publication.

Author Response

Thank you for your kind comments on our paper.

Reviewer 2 Report

The authors have improved the paper significantly. They have added additional trials and conducted further analysis, which has resulted the original conclusions to change slightly.

Few minor corrections are suggested:

Figure 2: Please include the citation as a reference number, not as reference link. Table 1: no need to show all the details of the trial. Presenting the statistics of the trials together with the number of trials mentioned in the text or the table should be sufficient. Comparison of the results of their work compared to Vicon system: better to show error if available rather than just saying "lower" or "comparable". Exact error figures are recommended.

Author Response

Thank you for your comments on our paper. We have been able to incorporate changes to reflect most of the suggestions. We have highlighted the changes in the red color within the manuscript. We hope that you find our responses satisfactory and that the paper is now acceptable for publication.

Comments and Suggestions for Authors

Few minor corrections are suggested:

Figure 2: Please include the citation as a reference number, not as reference link.

(We agree with this comment. Therefore, we have added a reference number. This change can be found in line number 166. )

Table 1: no need to show all the details of the trial. Presenting the statistics of the trials together with the number of trials mentioned in the text or the table should be sufficient.

(We agree with this comment. Therefore, we have revised Table 1. This change can be found in line numbers 353-356. )

Comparison of the results of their work compared to Vicon system: better to show error if available rather than just saying "lower" or "comparable". Exact error figures are recommended.

(We agree with this comment. Therefore, Figure 8 was added for comparison. This change can be found in line numbers 389-402 and 408-410. )